# Efficiency of *Rph genes* against *Puccinia hordei* in Southern Russia in 2019–2021

**Anastasia Danilova** * and **Galina Volkova** *

Federal State Budgetary Scientific Institution "Federal Research Center of Biological Plant Protection" (FSBSI FRCBPP), 350039 Krasnodar, Russia
* Correspondence: starlight001@yandex.ru (A.D.); galvol.bpp@yandex.ru (G.V.)

**Abstract:** Barley leaf rust (*Puccinia hordei* Otth.) is considered a harmful disease that occurs in barley-growing regions worldwide. In Russia, the disease is among the most prevalent in the Krasnodar region, which is the leader in the production of barley grain and has a favorable climate for disease development. In this paper, we studied the efficiency of 17 varieties and lines of barley from the International and Australian sets containing currently known *Rph* resistance genes or their combinations to *P. hordei* in the field, and 15 varieties and lines in the seedling phase in greenhouse conditions during 2019–2021. We concluded that the lines carrying the *Rph7* and *Rph13* genes remained immune throughout the three years of studies in the seedling and adult plant stages. The *Rph1* and *Rph23* genes showed moderate efficiency during the three years. The *Rph2, Rph3, Rph4, Rph5, Rph6+2, Rph8, Rph12, Rph19,* and *Rph21+2* genes showed low efficiency over the three years. This was also confirmed by the results of their assessment in the seedling phase: the number of monopustular isolates virulent to lines with the majority of the studied genes for three years was above 90%. Fluctuations in the virulence of the *P. hordei* population were observed under sufficiently unfavorable weather for disease development in 2019, 2020, and 2021. This proves the ability of the fungus to adapt to changing conditions. Therefore, annual monitoring of the response of lines and varieties carrying resistance genes and studying the virulence of the pathogen are crucial for the selection of rust-resistant varieties, and, hence, the prevention of barley leaf rust epidemics in all grain-producing regions worldwide.

**Keywords:** *Hordeum vulgare*; *Puccinia hordei*; *Rph* genes; population virulence; adult plant resistance; seedling plant resistance



## 1. Introduction

Barley (*Hordeum vulgare* subsp. *vulgare* L.) is the fourth most important cereal crop in the world after wheat, corn, and rice. The volume of production is 150 million tons; the area under crops is about 50 million hectares. The Russian Federation is the world leader in barley production, with a gross harvest, on average, that reaches 20 million tons per year [1]. The Krasnodar region ranks first among all regions of Russia in terms of barley sown area. For 2022, 179 thousand hectares of sown areas for barley were reserved in this region [2].

In the Krasnodar region, a large number of local barley varieties with different characteristics are cultivated. The size and quality of the grain harvest are under constant threat due to pathogenic fungi, and barley leaf rust is among these threats (causative agent—*Puccinia hordei* Otth.). Disease development on susceptible varieties can reach 60–80% under favorable climatic conditions [3]. Leaf rust is one of the most harmful and widespread foliar diseases of barley [4]. It is common in all areas of barley cultivation and can cause yield losses of up to 30% in susceptible varieties [5,6]. The disease leads to the formation of shriveled grain and a decreased yield. In infected plants, the efficiency of accumulation of organic matter decreases, the formation of pigments (chlorophyll and

carotenoids) is suppressed, the amount of storage carbohydrates (starch and sugar) in tissues decreases, and the activity of a number of enzymes is disturbed [7,8]. Optimum weather conditions, the constant presence of the host plant, the insufficient number of resistant varieties, and the monocropping of small grain cereals contribute to severe damages and extensive spread of the pathogen [9].

Currently, there is a pressing need for ecofriendly approaches to crop protection. The leading role, here, belongs to the selection and distribution of highly productive varieties that are resistant to diseases common in a certain agroclimatic zone. Genetic resistance serves as the best disease control measure as it is economical and environmentally sustainable and reduces dependence on fungicides.

N. I. Vavilov believed that the resistance of some crops to diseases is provided by a relatively small number of genes and is a consequence of the confrontation of two genotypes—a plant and a pathogen as a result of their joint evolution. Pathogens are constantly changing and overcoming the resistance I genes of the host plant [10–12].

To date, from the world literature, 23 genes are known that are responsible for seedling resistance (ASR): *Rph1–Rph19*, *Rph21*, *Rph22*, *Rph25*, and *Rph28*. Three other genes, in addition, are responsible for adult plant resistance (APR): *Rph20*, *Rph23*, and *Rph24*. All of these genes are found in cultivated barley (*H. vulgare*), wild barley (*Hordeum spontaneum* K. Koch), and bulbous barley (*Hordeum bulbosum* L.) [4,13–16]. Cultivars with APR genes are usually susceptible at the seedling stage, but at the adult stage, they slow down the development of the disease (slow rusting effect), prolonging plant resistance [4]. In addition to the main genes, small genes (quantitative trait loci, QTL) have been identified in barley, which can control a fairly high level of resistance to *P. hordei* [17].

The successful selection and pyramiding of the *Rph* genes of barley leaf rust require knowledge of their effectiveness at various phases of plant development. These studies should be carried out regularly.

Rogozhina E. M. (All-Russian Research Institute of Phytopathology) pioneered, in her published research results, *P. hordei* racial composition [18]. The N. I. Vavilov Institute of Plant Genetic Resources (VIR) undertook subsequent studies on the racial diversity of various geographical populations of *P. hordei* [19]. Some further endeavors on the study of the influence of environmental factors on the virulence and aggressiveness of the barley leaf rust were discussed by Tyryshkin [20].

Scientists from VIR in recent years have searched for sources of resistance among barley varieties to *P. hordei* [21–25]. Several other scientists have also carried out similar studies [26,27].

Since 2012, the Federal Research Center for Biological Plant Protection (FRCBPP) has begun research on the pathosystem in "barley—the causative agent of leaf rust". As a result, new highly virulent pathotypes of the fungus were reported [3]. It should be noted that such studies are regularly carried out in Russia in this scientific center alone. The FRCBPP constantly conducts research on the structure of the *P. hordei* population collected in the south of Russia. The center also evaluates the effectiveness of known *Rph* genes in the field of artificial infections with a pathogen.

Annual monitoring of the deployment of the known host plant R genes and their use in the cultivation of new barley varieties is extremely important. It is necessary to track annual changes in the virulence of the pathogen population and evaluate the efficiency of *Rph* genes. This is due to the fact that the population of *P. hordei* is changing. The pathogen population is influenced by a number of factors: the weather conditions of the growing season, the varieties sown in the region, the use of fungicides, the introduction of infection from other regions, etc. Due to active morphogenetic processes in the fungal population, such studies should be carried out continuously. This is the hypothesis and scientific novelty of the work.

In this paper, we aim to evaluate the efficiency of *Rph* genes in the adult and seedling stages in Southern Russia for the period 2019–2021.

## 2. Materials and Methods

### 2.1. 2019–2021 Growing Seasons

Seasonal weather of the three most significant months for the development of the disease (April, May, June) during 2019–2021 is shown in Figure 1. Data were provided by the weather station of the Federal Research Center of Biological Plant Protection, Krasnodar (FRCBPP).

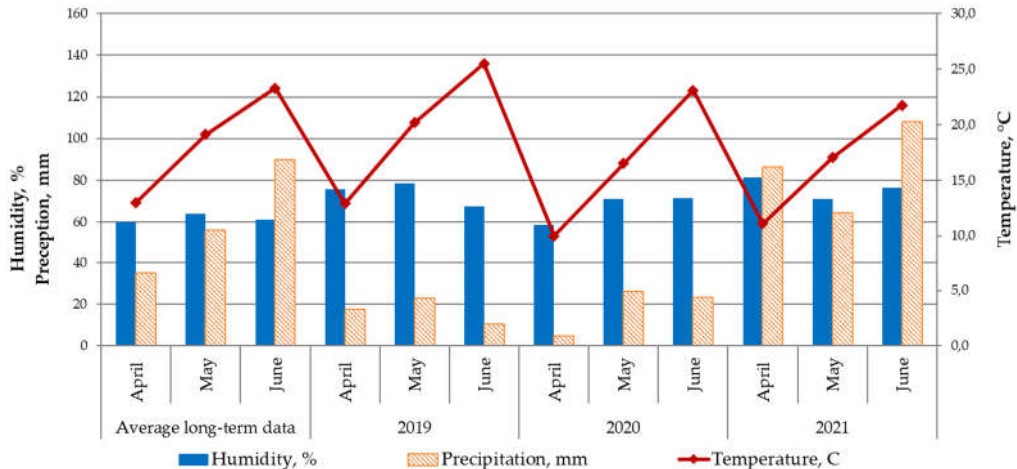

**Figure 1.** Climatogram of weather conditions for the research period 2019–2021 (according to the FRCBPP meteorological station, Russian Federation, Krasnodar region, Krasnodar).

The Krasnodar region is located in Southern Russia. This region is characterized by a temperate continental climate. The agroclimatic conditions of the growing seasons, depending on years of the research, were different. Autumn of 2018–2019 was cool, winter—warm and snowless. In the spring of 2019, sudden changes in air temperature did not contribute to the development of barley leaf rust. In April, a moderate temperature regime was noted with frosts in the air and on the soil surface and a lack of precipitation. May was distinguished by frequent heavy rainfall, which contributed to the disease.

The 2019–2020 seasons were characterized by a dry autumn. Winter was similar to the previous season. In the spring of 2020, in April and March, a lack of precipitation was observed, compared with the average long-term data. Heavy rainfall was noted in May. The 2020–2021 seasons were characterized by a dry autumn, a snowy winter, and a cold, long spring with heavy rains. The prevailing weather conditions could have influenced the development of *P. hordei* in the field experiment.

### 2.2. Obtaining Infectious Material

Infectious material, *P. hordei*, was obtained as a result of annual route inspections of production and breeding crops of winter barley in 2019–2021. In different years, between 18 and 26 districts of the Krasnodar region were surveyed. In each district, from 1 to 5 fields were surveyed. Collected barley leaves affected by leaf rust were signed and stored in filter paper at t 4–6 °C. The leaf samples were mixed to form a composite fungus population. Subsequently, the found infectious material was engrafted and multiplied on the susceptible line, L-94, in a greenhouse.

### 2.3. Efficiency Evaluation of Rph Genes in Adult Plants

Field trials were carried out at the infectious site of the FRCBPP (45.04822561304655 N, 38.87503073476379 E) from 2019 to 2021. We used 17 barley varieties from the International and Australian sets containing the currently known *Rph* genes or combinations thereof: *Rph1, Rph2, Rph3, Rph4, Rph5, Rph6+2, Rph7, Rph8, Rph9, Rph12, Rph13, Rph14, Rph19, Rph20, Rph21+2, Rph23+2,* and *Rph25*. The tested varieties were sown in single

rows of 1 m with row spacing of 0.5 m between the samples in 3 repetitions. A row of susceptible line L-94 was seeded through five test samples for accumulation and uniform distribution of *P. hordei* inoculum.

Plants were infected with *P. hordei* spores in the booting phase (Z-32). Fungal urediniospores were mixed with talc in a ratio of 1:100 at a load of 5 mg spores/m$^2$. The resulting mixture was used to pollinate the experimental plot in the evening after dew or light drizzle [28]. The development of barley leaf rust was assessed in the phase of milky-wax ripeness of the grain (Z-75–80)—Figure 2. The intensity of plant damage was determined by considering the area of the affected surface of leaves covered with pustules using a modified Cobb scale (Figure 3), characterizing the leaves from 0 to 100% infection covering the upper surface of the flag leaf, and corresponding the damage to the following classification: types of infection, where: O—no disease; R—resistance (instead of pustules, clearly defined spots of chlorosis are formed, leaf damage is up to 5–10%); MR—medium resistance (pustules are very small, surrounded by a chlorotic zone, leaf damage is not more than 10–30%); MS—medium susceptibility (small pustules, leaf damage up to 40–50%); S—susceptibility (large pustules, leaf damage from 50 to 75–100%). In each replicate, 10 plants were evaluated. The results of the accounts were summarized, after which the arithmetic mean for each option was calculated. Results have been rounded for presentation.

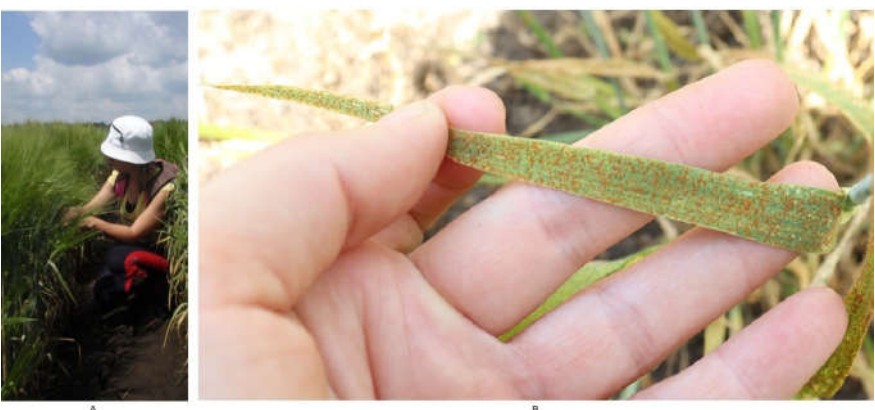

**Figure 2.** Accounting for the development of barley leaf rust in the field: (**A**) accounting for the development of the disease (orig.); (**B**) a barley leaf infected with a leaf rust pathogen (orig.).

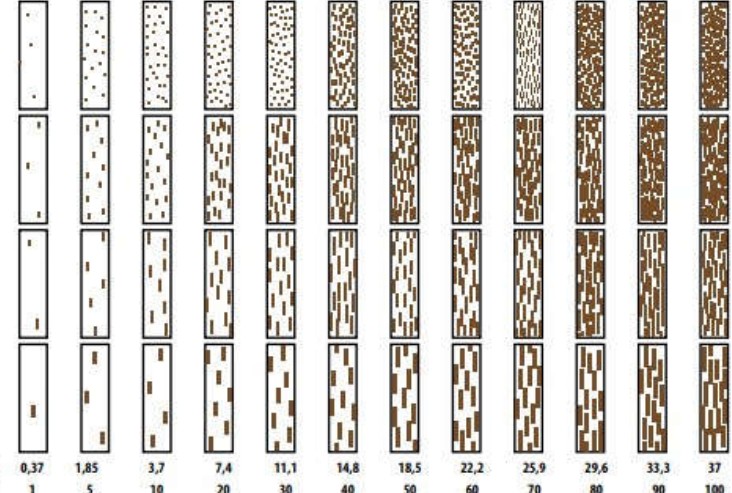

**Figure 3.** A scale for accounting for the infestation of cereals with types of rust (CIMMYT) [29,30]. Note: A—the actual area of the sheet covered with rust pustules, %; B—the degree of rust damage according to the modified Cobb scale.

*Rph* genes were ranked as follows: highly effective (plants without signs of damage), effective (leaf rust development 1R–5R), moderately effective (leaf rust development 10MR–20MR), and inefficient (leaf rust development 25 MS and above). Ranking was carried out according to the CIMMYT scale [31].

### 2.4. Efficiency Evaluation of Rph Genes in Seedling Plants

To assess the effectiveness of the *Rph* genes in the seedling phase, we used spore material of the leaf rust pathogen obtained as a result of annual route inspections of production and breeding crops of winter barley in 2019–2021. The material was grafted and multiplied on seedlings of the susceptible line, L-94. Inoculation of barley seedlings was carried out when the first leaf was fully opened. The wax coating was removed from the plants by rubbing the leaves with fingers moistened with water. *P. hordei* spore suspension was prepared in ceramic cups with the addition of a small amount of water. The leaf was infected by rubbing the urediniospore suspension onto the leaf. The plants were then sprayed with dew using a spray bottle and placed in humid chamber at a temperature of 18–20 °C for 16–24 h. After inoculation, the seedlings were placed in a separate chamber and kept under optimal conditions for the pathogen: temperature +18–22 °C; light intensity 12–15 thousand lux for 16 h. The incubation period was 5–7 days. Eight or ten days after inoculation, plants with a single pustule were separated with an isolator. L-94 seedlings were infected with spores obtained from a single pustule [28]. A total of 33, 37, and 58 monopustular isolates were obtained in 2019, 2020, and 2021, respectively.

Efficiency evaluation of *Rph* genes in the seedling stage was carried out in the greenhouse conditions of the FRCBPP in 2019–2021 on a set of barley varieties and lines containing *Rph* genes or their combinations: *Rph1*, *Rph2*, *Rph3*, *Rph4*, *Rph5*, *Rph6+2*, *Rph7*, *Rph8*, *Rph9*, *Rph12*, *Rph13*, *Rph14*, *Rph19*, *Rph21+2*, and *Rph25*. The seeds of each variety of the presented set were preliminarily germinated in Petri dishes and then planted with tweezers in plastic vases with a volume of 25 mL with wet sand, 5 seeds each. The seedlings were germinated for 7 days in isolated greenhouse boxes. Furthermore, sets of seedlings of varieties and lines with *Rph* genes were inoculated with each of the *P. hordei* monopustular isolates (Figure 4). On days 10–14, infectious types of differential reactions were assessed (in points) according to the Levin and Cherevik scale [32] (Figure 5).

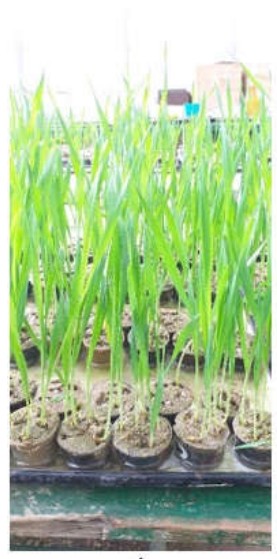
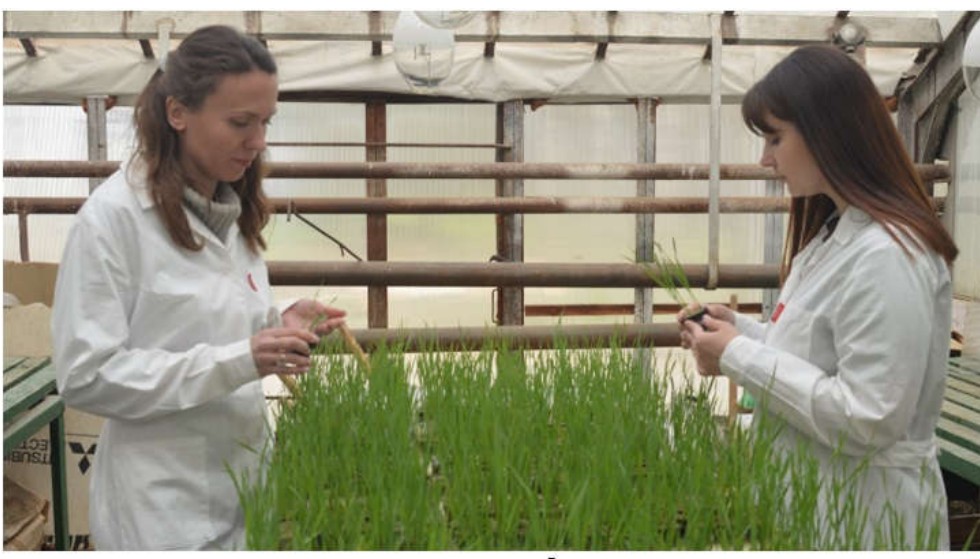

**Figure 4.** Accounting for the development of barley leaf rust in the greenhouse: (**A**) barley seedlings infected with monopustular isolates of *P. hordei* (orig.); (**B**) Accounting for the types of reactions of seedlings of barley differentiator varieties to infection with *P. hordei* monopustular isolates (orig.).

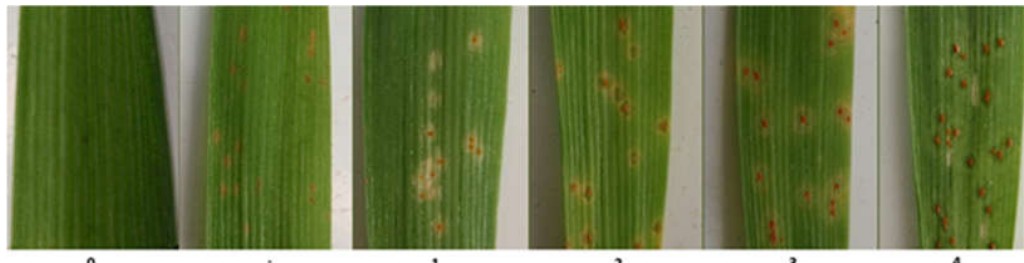

**Figure 5.** Range of seedling infection types for the *Puccinia hordei–Hordeum vulgare* interaction. The infection types were as follows: (**0**) = no visible symptoms; (**;**) = hypersensitive flecks; (**1**) = minute uredinia surrounded by mainly necrotic tissue; (**2**) = small to medium sized uredinia surrounded by chlorotic and/or necrotic tissue; (**3**) = medium to large uredinia with or without surrounding chlorosis; (**4**) = large uredinia without chlorosis. Infection types of 3+ or higher were considered to be compatible (i.e., virulent pathogen/susceptible host) [33,34].

The efficiency of *R* genes at the seedling stage was assessed by the frequency of virulent *P. hordei* isolates to varieties and lines with *Rph*. The percentage ratio of isolates virulent to one or more *Rph* genes to the total number of the studied isolates of the *P. hordei* was calculated.

## 3. Results

We have not identified any genes that showed absolute efficiency (there are no signs of damage on plants). The *Rph7* and *Rph13* genes proved to be highly efficient against *P. hordei*. During three years of research, lines with these genes had a minimal infection according to the reaction type R (according to the CIMMYT scale). The *Rph1* and *Rph23+2* genes showed moderate efficiency (10–20MR according to the CIMMYT scale) for three years. The *Rph14* and *Rph20* genes evaluated in 2021 also showed moderate efficacy. The *Rph6+2*, *Rph8*, *Rph12*, and *Rph19* genes were moderately effective in 2019, but have become ineffective in subsequent years. Their prevalence and type of response to pathogen infection for the period from 2019 to 2021 changed from 5–10MR to 30–60MS and 30–60S. The *Rph2*, *Rph3*, *Rph4, Rph5, Rph9,* and *Rph21+2* genes showed little efficiency. The development of *P. hordei* on these accessions ranged from 10–20MS to 40–60MS and 40–60S. The *Rph25 gene* studied in 2021 was ineffective as well (MS). Table 1 presents the results of the field evaluation of barley varieties and lines.

The effectiveness of *Rph* genes in the seedling stage against *P. hordei* isolates was evaluated under greenhouse conditions (Southern Russia, 2020–2021).

Monopustular isolates virulent to lines with the *Rph7* and *Rph13* genes were not detected in 2020–2021. There was a decrease in the effectiveness of the *Rph5* gene, since the number of monopustular isolates virulent to it had increased over the three years (from 48.6% to 94.8%). We registered the fluctuation of monopustular isolates virulent to the line with the *Rph3* gene; however, their number remained high (59.5–98.3%). A decrease in the number of isolates virulent to the line with the *Rph9* was revealed. For the *Rph14* gene, studied in 2021, a small number of virulent isolates (1.7%) were identified. The number of monopustular isolates virulent to lines with *Rph* genes *1, 8, 12,* and *21+2* was above 90% for two years, including the line with the *Rph25* gene, studied in 2021. The results are shown in Table 2.

**Table 1.** Immunological evaluation of barley varieties and lines with known R genes (*Rph*) to *P. hordei* in Southern Russia (FRCBPP infectious nursery, 2019–2021).

| *Rph* Gene(s) | Variety/Line | Disease Development, % and Plant Infection Types by Year | | |
| --- | --- | --- | --- | --- |
| | | 2019 | 2020 | 2021 |
| *Rph1* | Sudan | 1MR [1] | 1MR | 10MR |
| *Rph2* | Peruvian | 20MS | 40MS | 40MS |
| *Rph3* | Estate | 10MS | 10MS | 50S |
| *Rph4* | Gold | 10MS | 30MS | 40S |
| *Rph5* | Magnif 102 INTA | 25MS | 30MS | 50MS |
| *Rph6+2* | Bolivia | 5MR | 15MS | 60S |
| *Rph7* | Cebada Capa | 1R | 1R | 1R |
| *Rph8* | Egypt 4 | 10MR | 20MR | 30MS |
| *Rph9* | Abyssinian | 15MS | 15MS | 60S |
| *Rph12* | Trumpf | 5MR | 30MR | 40S |
| *Rph13* | PI 531849 | 1R | 1R | 5R |
| *Rph14* | PI 584760 | - ** | - | 20MR |
| *Rph19* | Prior | 10MR | 10MS | 40MS |
| *Rph20* | Vada | - | - | 10MR |
| *Rph21+2* | Ricardo | 25MS | 30MS | 60S |
| *Rph23+2* | Yerong | 5MR | 20MR | 20MR |
| *Rph25 (RphFT)* | Fong Tien | - | - | 15MS |
| absent | L-94 | 60S | 60S | 80S |

Note: [1] O—No disease; R—resistance (together with pustules, clearly defined chlorosis spots are formed, leaf damage is up to 5–10%); MR—medium resistance (pustules are very small, surrounded by a chlorotic zone, leaf damage is not more than 10–30%); MS—medium susceptibility (pustules are small, leaf damage is up to 40–50%); S—susceptibility (large pustules, leaf infestation from 50 to 75–100% ). **—No studies due to the lack of seed material.

**Table 2.** Frequency of virulent monopustular isolates in the *P. hordei* population in Southern Russia (greenhouse, FRCBPP, 2020–2021 ГГ.).

| *Rph* Gene(s) | Variety/Line | Frequency of Virulent Isolates, % | |
| --- | --- | --- | --- |
| | | 2020 | 2021 |
| *Rph1* | Sudan | 91.9 | 96.6 |
| *Rph2* | Peruvian | 73.0 | 96.6 |
| *Rph3* | Estate | 59.5 | 98.3 |
| *Rph4* | Gold | 89.2 | 93.1 |
| *Rph5* | Magnif 102 INTA | 48.6 | 94.8 |
| *Rph6+2* | Bolivia | 100 | 82.8 |
| *Rph7* | Cebada Capa | 0.0 | 0.0 |
| *Rph8* | Egypt 4 | 94.6 | 91.4 |
| *Rph9* | Abyssinian | 91.9 | 17.2 |
| *Rph12* | Trumpf | 91.9 | 94.8 |
| *Rph13* | PI 531849 | 0.0 | 0.0 |
| *Rph14* | PI 584760 | - [1] | 1.7 |
| *Rph19* | Prior | 27.0 | 32.8 |
| *Rph21+2* | Ricardo | 97.3 | 91.4 |
| *Rph25 (RphFT)* | Fong Tien | - | 94.8 |
| Number of isolates | - | 37 | 58 |

Note: [1]—No studies due to the lack of seed material.

## 4. Discussion

In our studies, only the *Rph7* and *Rph13* genes were resistant against the *P. hordei*: lines carrying these genes remained immune during the three years of research in all phases of plant development. The *Rph1*, *Rph14*, *Rph20*, and *Rph23* genes showed moderate efficiency in the field (10MR–20MR on the CYMMIT scale).

A decrease in the number of monopustular isolates virulent to lines with the *Rph9* and *Rph19* was noted, especially in comparison to studies of previous years [34]. The *Rph14*

has a low content of virulent isolates in 2021. This makes it possible to predict the high efficiency of this gene in the future. The efficiency of most genes decreased throughout the study period. This may be due to the influence of weather conditions, which in 2021 were most favorable for the pathogen. In addition, the influence of other biotic and abiotic factors is also possible.

It is known from the literature that temperature can play an important role in the interaction between the host plant and the rust fungus. At high temperatures, individual genes are able to show better expression [35–38]. In our studies in 2019, the temperature in April, May, and June was close to or slightly above the long-term average. In 2020 and 2021, these values were slightly lower than the average long-term data by 2–3 °C. This, for example, can explain the higher efficiency of the *Rph2*, *Rph19*, and *Rph23+2* genes in 2019 and their lower efficiency in 2020 and 2021.

Precipitation in April, May, and early June is a serious factor in the development of barley leaf rust in the Krasnodar region. According to long-term average data, this period is characterized by heavy rainfall. In 2019 and 2020, there was a deficit of precipitation, which could have a negative impact on the development of the pathogen. In 2021, these months were characterized by heavy precipitation, with almost twice the long-term average. In addition, during this growing season, higher air humidity was observed. The combination of these factors provided a favorable background for *P. hordei* in all studied varieties and lines with *Rph* genes.

A comparison of the obtained results with the data of other researchers indicates a difference in the effectiveness of a number of *Rph* genes [34].

The *Rph13* gene is noted in the literature as moderately effective [20], while in our studies this gene showed high efficiency for three years. In Europe and Australia, this gene is currently ineffective [4,8].

The *Rph20* gene, which has been shown to be moderately resistant, remains effective worldwide despite being used by breeders for many years. The recessive APR *Rph23* gene confers a very low level of resistance, which our studies have confirmed. Still, the literature review suggests that this gene, in combination with other APR genes such as *Rph20* (conferring moderate resistance), provides high protection against the pathogen [39,40].

In Australia, *Rph7*, *Rph14*, and *Rph21* are still effective. *Rph11*, *Rph15*, *Rph18*, and *Rph20* are also currently considered effective [4,41,42]. In our studies, *Rph21+2* showed low efficiency, and *Rph14* was characterized as moderately effective in 2021.

Previously, it was believed that the genes *Rph3*, *Rph7*, and, partly, *Rph9* are effective against barley leaf rust throughout the former USSR [43]. Subsequently, the researchers of VIR discovered high efficiency in only the *Rph7* gene in Russia [44]. Nevertheless, they also noted individual pathotypes virulent to this gene [45]. It is known from the literature that *Rph7*, previously highly effective against *P. hordei*, is losing its effectiveness worldwide. For example, this gene was considered effective in Europe for 20 years, but with the advent of new *P. hordei* pathotypes, it lost its effectiveness [46–49]. In the USA, *Rph7* has been found in most production varieties for 30 years. In the 1990s, with the advent of pathotypes virulent to this gene, strong epiphytoties of *P. hordei* were noted, as a result of which the efficiency of *Rph7* decreased significantly [5,48]. In some years, *Rph7* virulent pathotypes have been occasionally observed in Israel [40], Morocco [47], and North America [48].

In our studies in earlier periods, *Rph7* showed low efficiency, and, in some years, a high number of *P. hordei* isolates virulent to the line with this gene was noted [9]. However, starting from 2018, the number of isolates virulent to this line began to decrease. There was also a decrease in the number of isolates virulent to the line with *Rph9*. This is, perhaps, due to the fact that in earlier years the population brought from Asia or Europe prevailed in the territory of the south of Russia. Subsequently, the population of *P. hordei* was influenced by biotic and abiotic factors, such as the genotypes of barley varieties sown in the region, the use of fungicides, weather conditions, etc. These factors probably contributed to the selection of virulent isolates of the fungus. Nevertheless, the population of *P. hordei* in the south of Russia was characterized by high virulence to most of the known genes [9,34].

In general, to date, almost all identified genes have lost their effectiveness due to the emergence of new virulent *P. hordei* pathotypes [4]. In practice, three genes are known that are effective against the barley leaf rust pathogen: *Rph20*, *Rph23*, and *Rph24* [39,50,51]. Based on the results of our studies, we conclude that only the *Rph7* and *Rph13* genes are highly effective against the southern Russian population of *P. hordei* both in the seedling and adult phases. The APR genes *Rph20* and *Rph23+2* also proved to be effective.

The obtained results indicate the need for both monitoring the virulence of the barley leaf rust pathogen population and studying the efficiency of known *Rph* genes at different stages of plant development. This will help to correct the direction of plant genetic protection against the pathogen and make it more effective both in the south of Russia and in the world.

## 5. Conclusions

We studied the efficiency of *Rph* genes against barley leaf rust pathogen in seedling and adult plants in 2019–2021 in Southern Russia. The research results prove that the vast majority of known *Rph* genes are ineffective against the barley leaf rust pathogen. The information obtained should be taken into account when cultivating barley. In addition, there are still few data on the genetics of resistance of barley varieties in southern Russia. We intend to conduct such a study in the future.

As a result of the loss of efficiency of *Rph* genes in the field and in the seedling phase, a more thorough study of the genetics of resistance of widely sown varieties using various methods is necessary. Annual monitoring of the response of varieties and lines with *Rph* genes and the study of pathogen virulence are important measures necessary to prevent barley leaf rust epiphytoties in all grain-producing regions of the world.

**Author Contributions:** Conceptualization, A.D. and G.V.; research, A.D.; article writing—initial preparation of the project, A.D.; writing—review and editing, A.D. and G.V.; project administration, G.V. All authors have read and agreed to the published version of the manuscript.

**Funding:** The research was carried out in accordance with the State task of the Ministry of Science and Higher Education of the Russian Federation within the framework of research on topic No. FGRN-2022-0004. The study used the infrastructure and objects of the "State collection of entomoacariphages and microorganisms" (https://ckp-rf.ru/catalog/usu/585858/ (accessed on 17 February 2022)) and "Phytotron for the isolation, identification, study and maintenance of races, strains, phenotypes of pathogens" (https://ckp-rf.ru/catalog/usu/671925/ (accessed on 17 February 2022)).

**Institutional Review Board Statement:** Not Applicable.

**Informed Consent Statement:** Informed consent was obtained from all subjects involved in the study.

**Data Availability Statement:** All data obtained is contained in this article.

**Acknowledgments:** The authors are grateful to Olga Fedorovna Vaganova, Olga Alexandrovna Kudinova, Irina Petrovna Matveeva, and all the staff of the Laboratory of Plant Immunity to Diseases for their invaluable help in the research.

**Conflicts of Interest:** The authors declare no conflict of interest.

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
