# Peer review of "Efficiency of Rph genes against Puccinia hordei in Southern Russia in 2019–2021"

_agronomy, doi:10.3390/agronomy13041046_

Round 1

Reviewer 1 Report

Pictures during experiment in the field and in laboratory should be added in the manuscript. 

Author Response

- Thank you for your positive response. We have added some photos taken during research in the field and greenhouse.

Reviewer 2 Report

The paper presents interesting  results.

Manuscript is well structured and written in clear and concise manner.

In general, I do not have major remarks.

There are some minor issues that should be resolved:

Abstract - write in the past tense (i.e. we studied, we concluded)

Results - The presentation of the results from the previous study (reference 34) should be put in a proper way into the discussion section and than compared with the ones from the present study. 

Author Response

- Thank you for your positive response. We have made edits to the manuscript that were recommended by you.

Abstract - write in the past tense (i.e. we studied, we concluded)

- Fixed

Results - The presentation of the results from the previous study (reference 34) should be put in a proper way into the discussion section and then compared with the ones from the present study. 

- Fixed

Reviewer 3 Report

1.     In this manuscript, there were a few tense typos. Authors should have a check and correct carefully.

For example,‘is’ (line 16) should be ‘was’. ‘is’ (line 94) should be ‘was’.

2.     For references, there were a few kinds of mistakes in your manuscript.

For journals, the form should be ‘Author 1, A.B.; Author 2, C.D. Title of the article. Abbreviated Journal Name YearVolume, page range.’ The journal name should be abbreciated and italic, and ‘;’ was used between two authors’ name. At the end of references, there is no need to display the doi numbers. Authors should correct all the references in this manuscript following with instructions carefully (Agronomy | Instructions for Authors (mdpi.com)).

3.     For introduction part, there was only a few basic information of barley and barley leaf rust, without any research progress about Rph genes in barley and any other species. The  problems encountered in recently research or lack of research were not be listed. Some sentences were not useful and some sentences only mentioned research basis about barley leaf rust. There is no explanation of the relationship between these works and this study or what scientific issues that need to be further studied. In addition, this section had too many paragraphs without good induction and summary. Authors also didn’t explain the significance of your research. At the end of this section, there should be one paragraph gave us some information about experimental content, main methods and the key problems to be solved. As far as I’m concerned, this section should be rewritten.

4.     For materials & methods, the presentation was too complicated and needed to be simplified appropriately. Firstly, the subtitle of this part all had the wrong number. For example, the ‘4.1’ should be ‘2.1’. For the first part, it didn’t need to use three paragraphs for describing the weather condition, only a few sentences were enough. There was a figure displayed the weather data of 2019-2021. In fact, this figure can be put in the supplementary files. For the third paragraph, there was lack of damage standard figures for barley leaf rust. For the last part, the calculating methods for Rph genes efficiency was missing.

5.     For results, all the tables were not standard and the notes were not clear. ‘*’ was usually used as significance but not number. And there was lack of instructions for numbers in table 1.

6.     In this article, I think there should be added some data and experiments. In the abstract, you mentioned the seedling and adult plant age, but there were lack of data for seedlings’ resistance, and also without any data for describing the relationship between seedling and adult plant age. Secondly, the weather data was displayed in the methods and materials part. Maybe you can design an experiment to research the relationship between seedling resistance and condition indicators, and also the interaction among these three indicators. Thirdly, you only counted the efficiency, but didn’t mention the growth and physiological situation. The expressions of Rph genes were also missing, and seedling status before and after infection were also lack of data and figures support.

7.     In discussion, the indicator analysis of weather was lack of data support.

8.     At the end of discussion part, the line was not necessary.

9.     In conclusions, the expression was not concise enough.

Author Response

- Thank you for such a thorough analysis of the manuscript. We have made edits to the manuscript that were recommended by you and prepared responses to comments.

  1. In this manuscript, there were a few tense typos. Authors should have a check and correct carefully. For example,‘is’ (line 16) should be ‘was’. ‘is’ (line 94) should be ‘was’.

- Fixed

  1. For references, there were a few kinds of mistakes in your manuscript. For journals, the form should be ‘Author 1, A.B.; Author 2, C.D. Title of the article. Abbreviated Journal Name Year, Volume, page range.’ The journal name should be abbreciated and italic, and ‘;’ was used between two authors’ name. At the end of references, there is no need to display the doi numbers. Authors should correct all the references in this manuscript following with instructions carefully (Agronomy | Instructions for Authors (mdpi.com)).

- Fixed

  1. For introduction part, there was only a few basic information of barley and barley leaf rust, without any research progress about Rph genes in barley and any other species. The problems encountered in recently research or lack of research were not be listed. Some sentences were not useful and some sentences only mentioned research basis about barley leaf rust. There is no explanation of the relationship between these works and this study or what scientific issues that need to be further studied. In addition, this section had too many paragraphs without good induction and summary. Authors also didn’t explain the significance of your research. At the end of this section, there should be one paragraph gave us some information about experimental content, main methods and the key problems to be solved. As far as I’m concerned, this section should be rewritten.

- Added some suggestions about the importance of ongoing research.

  1. For materials & methods, the presentation was too complicated and needed to be simplified appropriately. Firstly, the subtitle of this part all had the wrong number. For example, the ‘4.1’ should be ‘2.1’. For the first part, it didn’t need to use three paragraphs for describing the weather condition, only a few sentences were enough. There was a figure displayed the weather data of 2019-2021. In fact, this figure can be put in the supplementary files. For the third paragraph, there was lack of damage standard figures for barley leaf rust. For the last part, the calculating methods for Rph genes efficiency was missing.

- Numbering of subtitles has been corrected.

A detailed description of the weather conditions was made so that later the results could be explained. The development of the pathogen of dwarf rust depends on this factor. It's also affects the virulence of the fungus population. In addition, some Rph genes are more efficient, depending on the temperature during the growing season.

Scales have been added to the section to account for the development of rust on adult plants and types of reaction on seedlings.

The calculation of the efficiency of Rph genes is described in the section. Added a description of the calculation of the effectiveness of Rph-genes in the seedling phase.

  1. For results, all the tables were not standard and the notes were not clear. ‘*’ was usually used as significance but not number. And there was lack of instructions for numbers in table 1.

- Fixed

  1. In this article, I think there should be added some data and experiments. In the abstract, you mentioned the seedling and adult plant age, but there were lack of data for seedlings’ resistance, and also without any data for describing the relationship between seedling and adult plant age. Secondly, the weather data was displayed in the methods and materials part. Maybe you can design an experiment to research the relationship between seedling resistance and condition indicators, and also the interaction among these three indicators. Thirdly, you only counted the efficiency, but didn’t mention the growth and physiological situation. The expressions of Rph genes were also missing, and seedling status before and after infection were also lack of data and figures support.

- The relationship between the efficiency of Rph-genes in different phases of development can be traced in some cases, while not in others. In the «discussion» section, there was an attempt to explain the reason for this.

  1. In discussion, the indicator analysis of weather was lack of data support.
  2. В ходе обсуждения анализ показателей погоды был связан с отсутствием поддержки данными.

- The article was an attempt to explain the results obtained as a result of assessing the efficiency of Rph-genes in the phase of an adult plant, the prevailing weather conditions in the growing season.

  1. At the end of discussion part, the line was not necessary.

- Fixed

  1. In conclusions, the expression was not concise enough.

- Fixed. The section «Conclusions» has been shortened. Part of the text from this section has been moved to the «Discussions» section.

Reviewer 4 Report

In this manuscript, the efficiency of Rph genes against Puccinia hordei in Southern Russia was investigated in 2019-2021. This is a basic research work. Some comments are as follows.

1. What are the main pathotypes of the pathogen in Southern Russia? This can be described in the manuscript.

2. What are the growing areas of varieties containing the Rph genes resistant against the P. hordei population? This can be described in the manuscript.

3. In the Conclusions section, there are too many paragraphs, and some contents can be moved into the Discussion section.

4. In Line 12, please pay attention to the format of ‘P. hordei’.

5. In Line 58, please present the full name of ‘APR’.

6. In Line 92, 4.1 should be 2.1? The following 4.2?4.3?4.4? These should be corrected.

7. In Lines 119-120, ‘were signed and stored in filter paper at t 4-6 . Leaf samples were mixed to form a composite fungus population.’, the ‘.’ before ‘Leaf ’ disappeared. Please check it.

8. In Line 186, ‘10MR-20MR’? What's the meaning of it? This should be pointed out in the text.

9. In Lines 216-217, ‘The number of monopustular isolates virulent to lines with the Rph genes: 1, 2, 4, 6+2, 8, 12, 21+2 was above 90% for three years’, please check the data.

10. In Line 229, ‘2-3 C’? Please check it.

Author Response

- Thank you for your positive response. We have made edits to the manuscript that were recommended by you and prepared responses to comments.

  1. What are the main pathotypes of the pathogen in Southern Russia? This can be described in the manuscript.

- According to our studies, isolates virulent to the majority of Rph-genes predominate in the population of barley leaf rust pathogen in southern Russia; except for the Rph7 and Rph13. This information is mentioned in the article.

However, the main focus of this manuscript is to evaluate the efficacy of Rph-genes against the pathogen. We do not focus on the barley leaf rust population structure in this article.

  1. What are the growing areas of varieties containing the Rph genes resistant against the P. hordei population? This can be described in the manuscript.

- The barley samples used in our experiment are varieties and lines from different sources (different parts of the world). They have the Rph genes identified. These varieties and lines are used by most researchers to determine the virulence of the barley leaf rust pathogen. These accessions are often used in breeding programs.

For example, a barley accession containing the Rph13 gene (PI 531849) is an experimental line from the Cambridge Institute of Plant Breeding (UK) that was selected from a cross having an accession of Hordeum vulgare spontaneum from Israel as the donor parent. A sample carrying the Rph14 (PI 584760) comes from Egypt (Jin, 1994). The Cebada Capa variety (Rph7) was obtained in the USA. This gene has been used in many breeding programs in Europe and the USA. Rph20 and Rph23 have been postulated by a number of researchers in many Australian barley varieties (Park, 2015).

We cannot say which Rph genes are contained in barley varieties bred in the South of Russia. Their postulation is the subject of our future research.

Jin, Y., and B.J. Steffenson. 1994. Inheritance of resistance to Puccinia hordei in cultivated and wild barley. J. Hered. 85:451-454.

Park, R.F.; Golegaonkar, P.G.; Derevnina, L.; Sandhu, K.S.; Karaoglu, H.; Elmansour, H.M.; Dracatos, P.M.; Singh, D. Leaf Rust of Cultivated Barley: Pathology and Control. Annu. Rev. Phytopathol. 2015, 53, 565-589.

  1. In the Conclusions section, there are too many paragraphs, and some contents can be moved into the Discussion section.

- Fixed. Part of the text from the «Conclusions» section has been moved to the «Discussions» section.

  1. In Line 12, please pay attention to the format of ‘P. hordei’.

- Fixed

  1. In Line 58, please present the full name of ‘APR’.

- Fixed

  1. In Line 92, 4.1 should be 2.1? The following 4.2?4.3?4.4? These should be corrected.

- Fixed

  1. In Lines 119-120, ‘were signed and stored in filter paper at t 4-6 ℃. Leaf samples were mixed to form a composite fungus population.’, the ‘.’ before ‘Leaf ’ disappeared. Please check it.

- Fixed

  1. In Line 186, ‘10MR-20MR’? What's the meaning of it? This should be pointed out in the text.

- Fixed

  1. In Lines 216-217, ‘The number of monopustular isolates virulent to lines with the Rph genes: 1, 2, 4, 6+2, 8, 12, 21+2 was above 90% for three years’, please check the data.

- Fixed

  1. In Line 229, ‘2-3 C’? Please check it.

- Fixed

Round 2

Reviewer 3 Report

Though authors have revised some contents in the  manuscript of this version, there still some problems without being solved. In my opinion, this manuscript was  lack of some other experiments as I have mentioned last time and still need to be further completed. 

Author Response

Dear reviewer!
May we ask you again to provide detailed comments on our manuscript, taking into account the changes that we have already made?